# Enrichment of infection-associated bacteria in the low biomass brain bacteriota of Alzheimer's disease patients

Yeon Kyeong Ko[1], Eunbi Kim[1], Eun-Jae Lee[2], Soo Jeong Nam[3], Yeshin Kim◉[4], Seongheon Kim[4], Se-Young Choi[5], Hyun Young Kim[1], Youngnim Choi◉[1]*

1 Department of Immunology and Molecular Microbiology, School of Dentistry and Dental Research Institute, Seoul National University, Seoul, Republic of Korea, 2 Department of Neurology, Asan Medical Center, University of Ulsan College of Medicine, Seoul, Republic of Korea, 3 Department of Pathology, Asan Medical Center, Seoul, Republic of Korea, 4 Department of Neurology, Kangwon National University Hospital, Kangwon National University College of Medicine, Chuncheon, Republic of Korea, 5 Department of Physiology, Dental Research Institute, Seoul National University School of Dentistry, Seoul, Republic of Korea

* youngnim@snu.ac.kr

**Data Availability Statement:** The sequence data are available in the NCBI Sequence Read Archive under the accession number PRJNA1013214.

## Abstract

Alzheimer's disease (AD) is a neurodegenerative disease accompanied by neuroimmune inflammation in the frontal cortex and hippocampus. Recently, the presence of bacteria in AD-affected brains has been documented, prompting speculation about their potential role in AD-associated neuroinflammation. However, the characterization of bacteriota in human brains affected by AD remains inconclusive. This study aimed to investigate potential associations between specific bacteria and AD pathology by examining brain tissues from AD-associated neurodegenerative regions (frontal cortex and hippocampus) and the non-AD-associated hypothalamus. Employing 16S rRNA gene sequencing, 30 postmortem brain tissue samples from four individuals with normal brain histology (N) and four AD patients were analyzed, along with three blank controls. A remarkably low biomass characterized the brain bacteriota, with their overall structures delineated primarily by brain regions rather than the presence of AD. While most analyzed parameters exhibited no significant distinction in the brain bacteriota between the N and AD groups, the unique detection of *Cloacibacterium normanense* in the AD-associated neurodegenerative regions stood out. Additionally, infection-associated bacteria, as opposed to periodontal pathogens, were notably enriched in AD brains. This study's findings provide valuable insights into potential link between bacterial infection and neuroinflammation in AD.

## Introduction

Alzheimer's disease (AD) is a progressive neurodegenerative condition characterized by hippocampal and cerebral cortex shrinkage, leading to memory loss, language problems, and behavioral changes [1]. In the initial stage, AD manifests as mild cognitive decline and short-

**Funding:** This study was supported by the National Research Foundation of Korea (Daejun, Korea) through the grants 2018R1A5A2024418 and 2020R1A2C2007038 awarded to Youngnim Choi. The funders had no role in study design, data collection and analysis, decision to publish, or preparation of the manuscript.

**Competing interests:** The authors have declared that no competing interests exist.

term memory loss, progressing to severe memory impairment, apraxia, and aphasia [1–3]. AD is believed to be a multifactorial disease associated with various risk factors, including aging, genetic variants, obesity, diabetes, mitochondrial dysfunction, vascular diseases, immune system dysregulation, and infectious agents [4, 5].

Mounting evidence suggests that increased neuroinflammation plays a significant role in the pathogenesis of AD, and bacterial infection has emerged as a potential contributor to neuroinflammation [6–8]. Numerous studies have reported potential links between bacterial infections and AD progression. The gut microbiota can influence blood brain barrier (BBB) permeability [9, 10]. Metabolites, amyloids, and lipopolysaccharides (LPS) produced by gut microbiota may influence neuroinflammation and activate signaling pathways associated with AD progression [11–13]. Furthermore, the role of oral microbiota in neuroinflammation has also been suggested. For instance, DNA from the periodontal pathogen *Porphyromonas gingivalis* and its virulence factors, such as LPS and gingipains, have been detected in brain tissues of AD patients [14, 15]. Furthermore, oral administration of *P. gingivalis* in mice resulted in increased production of $A\beta_{1-42}$ and proinflammatory cytokines in brain tissue, leading to neurotoxicity and memory impairment [15, 16].

Although precise pathways of bacterial invasion into brain tissue and their contributions to AD progression remain elusive, considerable efforts have been made to investigate AD-associated bacteria in brain tissue. A study using bacteria-specific immunofluorescence and PCR reported polymicrobial infections in AD brain tissue, but the results of immunofluorescence and PCR did not always coincide [17]. The results of two studies utilizing 16S rRNA gene sequencing analysis did not coincide each other, and both studies lacked blank controls or elimination of off-target amplicons [18, 19]. Consequently, the presence of the AD-associated bacteriota in brain tissue remains uncertain.

To address this issue, we conducted 16S rRNA gene sequencing analysis on 30 brain tissue samples from four AD patients and four control individuals, alongside three negative blank controls. By analyzing brain tissues from both AD-associated neurodegenerative regions (frontal cortex and hippocampus) and the hypothalamus as a non-AD-associated region, we aimed to explore whether specific bacteria are associated with AD pathology.

## Materials and methods

### Human brain tissues

This study was performed in accordance with the Helsinki Declaration after an approval from the Institutional Review Board at Seoul National University School of Dentistry (S-D20210021). We obtained a total of 30 samples of 2 mm × 2 mm × 2 mm sized frozen postmortem brain tissues from four control individuals with normal brain histology (N) and four AD patients with clear AD pathology from the Korean Brain Bank Network (Dong-gu, Daegu, Republic of Korea). The postmortem brain tissues in the Korean Brain Bank Network are archived with a written consent for future use before death, and additional consent was not needed for the current study. The tissue samples were procured from the bank between 2021 November 25th and 2022 July 14th along with information in Table 1. Authors did not have access to information that could identify individual participants during or after data collection. Normal brain tissues included seven frontal cortex, seven hippocampus, and one hypothalamus, whereas AD brain tissues included seven frontal cortex, five hippocampus, and three hypothalamus. Two adjacent blocks from the same individual were labeled as (1) and (2).

**Table 1. Sample information.**

| Sample ID | DNA conc (ng/μL) | Tissue site | Type | Year of autopsy | Age (yr) | Gender | PMI (h) | Cause of death | Forebrain weight (g) | Thal phase | Breaak stage | ADNC |
|---|---|---|---|---|---|---|---|---|---|---|---|---|
| N1-F | 112.9 | F | Normal | 2020 | 72 | Male | 2 | No pathological diagnosis | 1,580 | Phase 0 | Stage 0 | Not AD |
| N1-HC | 61.3 | HC | | | | | | | | | | |
| N1-HT | 74 | HT | | | | | | | | | | |
| N2-F (1) | 80.1 | F | Normal | 2021 | 50 | Male | 6.5 | Metastatic colorectal cancer | 1,550 | Phase 0 | Stage 0 | Not AD |
| N2-F (2) | 69.3 | F | | | | | | | | | | |
| N2-HC (1) | 48.2 | HC | | | | | | | | | | |
| N2-HC (2) | 43.6 | HC | | | | | | | | | | |
| N3-F (1) | 18 | F | Normal | 2018 | 52 | Male | 4.5 | No pathological diagnosis | 1,015 | Phase 0 | Stage 0 | Not AD |
| N3-F (2) | 9.2 | F | | | | | | | | | | |
| N3-HC (1) | 58.7 | HC | | | | | | | | | | |
| N3-HC (2) | 33.8 | HC | | | | | | | | | | |
| N4-F (1) | 128.3 | F | Normal | 2020 | 83 | Male | 4 | No pathological diagnosis | 1,037 | Phase 0 | Stage 0 | Not AD |
| N4-F (2) | 51.3 | F | | | | | | | | | | |
| N4-HC (1) | 59.4 | HC | | | | | | | | | | |
| N4-HC (2) | 49.6 | HC | | | | | | | | | | |
| AD1-F | 103.3 | F | AD | 2019 | 82 | Male | 4 | AD | 1,350 | Phase 5 | Stage 3 | High |
| AD1-HC | 94.8 | HC | | | | | | | | | | |
| AD1-HT | 56.8 | HT | | | | | | | | | | |
| AD2-F (1) | 193.8 | F | AD | 2018 | 57 | Female | 2.5 | AD | 750 | Phase 5 | Stage 6 | High |
| AD2-F (2) | 84.5 | F | | | | | | | | | | |
| AD3-F (1) | 68.9 | F | AD | 2021 | 80 | Male | 12.5 | AD | 1,390 | Phase 4 | Stage 6 | High |
| AD3-F (2) | 70.9 | F | | | | | | | | | | |
| AD3-HC (1) | 66.9 | HC | | | | | | | | | | |
| AD3-HC (2) | 40.5 | HC | | | | | | | | | | |
| AD4-F (1) | 74.5 | F | AD | 2021 | 76 | Female | 4 | AD + Synucleinopathy | 1,191 | Phase 4 | Stage 6 | High |
| AD4-F (2) | 47.5 | F | | | | | | | | | | |
| AD4-HC (1) | 53 | HC | | | | | | | | | | |
| AD4-HC (2) | 32.9 | HC | | | | | | | | | | |
| AD4-HT (1) | 39.7 | HT | | | | | | | | | | |
| AD4-HT (2) | 45.8 | HT | | | | | | | | | | |

PMI: Postmortem interval, ADNC: Alzheimer's Disease Neuropathologic Change, F: Frontal cortex, HC: Hippocampus, HT: Hypothalamus

## DNA extraction

The tissues were subjected to genomic DNA extraction using the DNeasy PowerSoil Pro Kits (Qiagen, 47014, Hilden, Germany) that includes a mechanical agitation step with glass beads. Extracted DNA was eluted with 50 μL DNA elution buffer. A260/280 ratio of all samples was between 1.8 and 2, and DNA concentrations varied from 9.2 to 187 ng/μL. In addition to these DNA samples, one negative control sample in which DNA extraction had been conducted without a tissue was included.

## Quantification of bacterial copy numbers in samples

Real-time quantitative PCR (qPCR) was performed in a total volume of 20 μL, including 2 μL of genomic DNA, 0.2 μM of each universal primer (27F) (AGTCACTGACGAGTTTGATCMTG GCTCAG) and 518R (CAGTGACTACWTTACCGCGGCTGCTGG), and 10 μL of SYBR using a

StepOnePlus™ Real-Time PCR System (Applied Biosystems™, 4376600, Waltham, MA, USA) with the cycling conditions: 4 min denaturation at 95˚C followed by 40 cycles of denaturation at 95˚C for 15 s, annealing at 60˚C for 15 s, and extension at 70˚C for 33 s. A standard curve generated using *P. gingivalis* genomic DNA template was used to calculate copy number.

## 16S rRNA sequencing

Bacteria 16S rRNA gene sequencing was performed by CJ Bioscience, Inc. (Jung-gu, Seoul, Republic of Korea). Briefly, the V3-V4 regions of the 16S rRNA gene were amplified using loci-specific primers, 341F (`CCTACGGGNGGCWGCAG`) and 806R (`GACTACHVGGGTATC TAATCC`) with overhang adapters (`TCGTCGGCAGCGTC–AGATGTGTATAAGAGACAG` for forward; `GTCTCGTGGGCTCGG–AGATGTGTATAAGAGACAG` for reverse). PCR was performed in a total of 25 μL, comprising 2 μL genomic DNA and 10 pmol of each primer using the following cycling condition: initial denaturation at 95˚C for 3 min followed by 25 cycles of denaturation at 95˚C for 30 s, annealing at 55˚C for 30 s, and extension at 72˚C for 30 s, and the final extension at 72˚C for 5 min. The amplicons were re-amplified using index primers that target terminal 14/15 nucleotides of the overhang adapters of primers used in the first PCR. The second PCR was performed in a total of 25 μL, comprising 2 μL of cleaned 1st PCR product and 10 pmol of each primer using the following cycling condition: initial denaturation at 95˚C for 3 min followed by 8 cycles of denaturation at 95˚C for 30 s, annealing at 55˚C for 30 s, and extension at 72˚C for 30 s, and the final extension at 72˚C for 5 min. Three negative controls, one DNA extraction control (NC1) and two no-template controls (NC2 and NC3) were included in the PCR and sequencing to exclude potential contaminants from the data. After clean-up and quantification, the PCR products were pooled and subjected to sequencing using Illumina MiSeq (Illumina, San Diego, CA, USA) sequencing system. The sequence data are available in the NCBI Sequence Read Archive under the accession number PRJNA1013214 (https://www.ncbi.nlm.nih.gov/sra/PRJNA1013214).

## Bacteriota analysis

Processing of raw reads was started with quality check and filtering of low quality ($<$ Q25) reads using Trimmomatic ver. 0.321. After QC pass, paired-end sequence data were merged together using fastq_mergepairs command of VSEARCH version 2.13.4 with default parameters. Primers were then trimmed with the alignment algorithm of Myers & Miller at a similarity cut-off of 0.8. Non-specific amplicons that do not encode 16S rRNA were detected by the program nhmmer in HMMER software package ver. 3.2.1 with hmm profiles. Unique reads were extracted and redundant reads were clustered with the unique reads by using the derep_- fulllength command of VSEARCH. The EzBioCloud 16S rRNA database was used for taxonomic assignment using the usearch_global command of VSEARCH followed by more precise pairwise alignment. Chimeric reads were filtered on reads with <97% similarity by reference-based chimeric detection using the UCHIME algorithm and the non-chimeric 16S rRNA database from EzBioCloud. After chimeric filtering, reads that were not identified to the species level (with <97% similarity) in the EzBioCloud database version PKSSU 4.0 (https://www.ezbiocloud.net/) were compiled and the cluster_fast command was used to perform de-novo clustering to generate additional operational taxonomic units (OTUs). Finally, OTUs with single reads (singletons) were omitted from further analysis. We analyzed the microbiome in the samples using the species that passed three steps of the filtering process (Fig 1A). Microbiome analysis of alpha- and beta-diversities were performed using the vegan package in R. Bray–Curtis dissimilarity was used to examine the overall phylogenetic distance between communities, which was visualized using multidimensional scaling (MDS).

### Bacterial source

We searched for the sources of the identified taxa using the information at the Bacterial and Viral Bioinformatics Resource Center (https://www.bv-brc.org/) and DSMZ (https://www.dsmz.de/). For information on most gut microbiota, a previous report was referred [20].

### Statistics

Differences in an intergroup phylogenic distance were analyzed by permutational multivariate analysis of variance (PERMANOVA) performed with Adonis function using the vegan package in R. To determine differences in alpha- and beta-diversities, relative abundance of each taxa, and bacterial sources, Mann–Whitney U test or the Kruskal–Wallis test was performed depending on the number of comparison groups using SPSS Statistics 26.0 software (IBM, Chicago, IL, USA). Statistical significance was set at $P < 0.05$, except for the differences in the relative abundance of each taxa where significance was set at $P < 0.01$. None of the significant taxa passed the Benjamini–Hochberg test to correct multiple comparisons.

## Results

### Strategy for AD-associated bacteriota analysis in human brain tissues

We hypothesized that AD-associated bacteria would be present or increased specifically in the AD-associated neurodegenerative regions (frontal cortex and hippocampus) of AD patients, while being absent in the same regions of control individuals (N) and in the non-AD-associated region (hypothalamus). We originally planned to obtain brain tissues from all three areas from each donor. However, due to limited tissue availability, we were only able to obtain brain tissues from all three areas for donors N1, AD1, and AD4. AD2 lacked hippocampus tissue as well (Table 1). To ensure the integrity of our analysis of low biomass microbial communities, we included three negative control samples (NC1-NC3) for validation purposes.

From the 16S rRNA sequencing, an average of 64,954 ± 22,151 total reads were obtained from the brain tissues, and 4,410 ± 5,007 total reads from the NCs. After filtering out non-specific amplicons, non-target sequences (archaea and eukarya), and chimera reads (Filter 1), an average of 2,290 ± 2,759 valid reads remained for the brain tissues, and 3,817 ± 4,223 valid reads for the NCs, with a taxonomic assignment to 1,585 OTUs at the species level. Notably, 85.6% to 99.8% of total reads were removed as non-specific amplicons in brain tissue samples, indicating off-target amplification of the host genome in these samples. In contrast, only 7.7% to 19.1% of reads were removed from the NCs. After applying Filter 2 to exclude low-abundance taxa (valid reads with ≤ 2 in each sample), a total of 961 OTUs remained. Finally, potential contaminants were removed using the exact binomial test (Filter 3), resulting in 710 OTUs being retained for subsequent analysis (Fig 1A). Although the number of the final valid reads obtained from each sample was quite small, the Good's coverage of library was higher than 95% except for five samples, namely, N1–HT, N1–F, N4–F (1), AD1–HT, AD1–F.

Interestingly, the number of valid reads were significantly higher in the hippocampus than in the frontal cortex or hypothalamus. However, no significant differences were observed between the N and AD groups, irrespective of whether the hypothalamus samples from AD patients were grouped as N or AD (Fig 1B and S1A Fig). Similarly, the copy numbers determined by qPCR trended to be higher in the hippocampus than in the frontal cortex or hypothalamus, although not statistically significant, and similar median values were observed between the N and AD groups (Fig 1C).

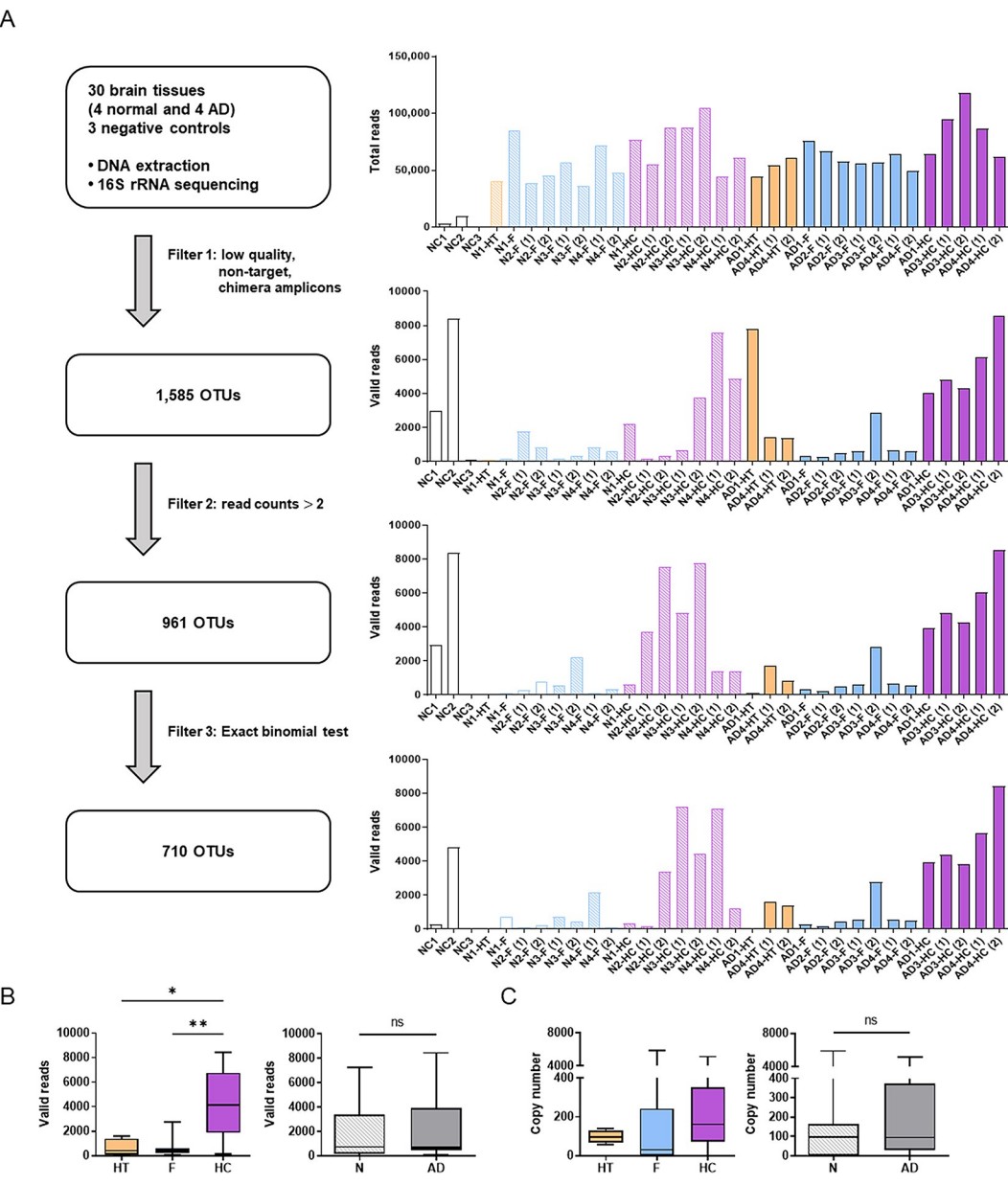

**Fig 1. Workflow of filtering steps of valid reads for microbiome analysis.** (A) Summary of filtering steps and the valid reads after each filter is presented. The 16S rRNA sequencing was performed on 30 human brain tissues from four normal individuals and four AD patients, and three negative controls. In the initial filtration phase, low-quality, off-target, and chimeric amplicons were eliminated from the dataset (Filter 1). Subsequently, species that displayed valid reads with $\leq 2$ across all samples were excluded (Filter 2). The final filtration step involved the application of an exact binomial test to discern potential contaminants. Any identified potential contaminant was subsequently removed from the dataset (Filter 3) (B) The valid reads and (C) the copy number from the Filter 3 depending on groups are illustrated as box and whisker plots. The significance among the HT, F, and HC and between N and AD was examined by the Kruskal–Wallis test and Mann–Whitney *U* test, respectively.

## Overall structures of brain bacteriota

We assessed alpha-diversities within the samples by evaluating the number of OTUs and the Shannon diversity index. Both parameters were significantly higher in the hippocampus than in the frontal cortex. However, there were no significant differences between the N and AD

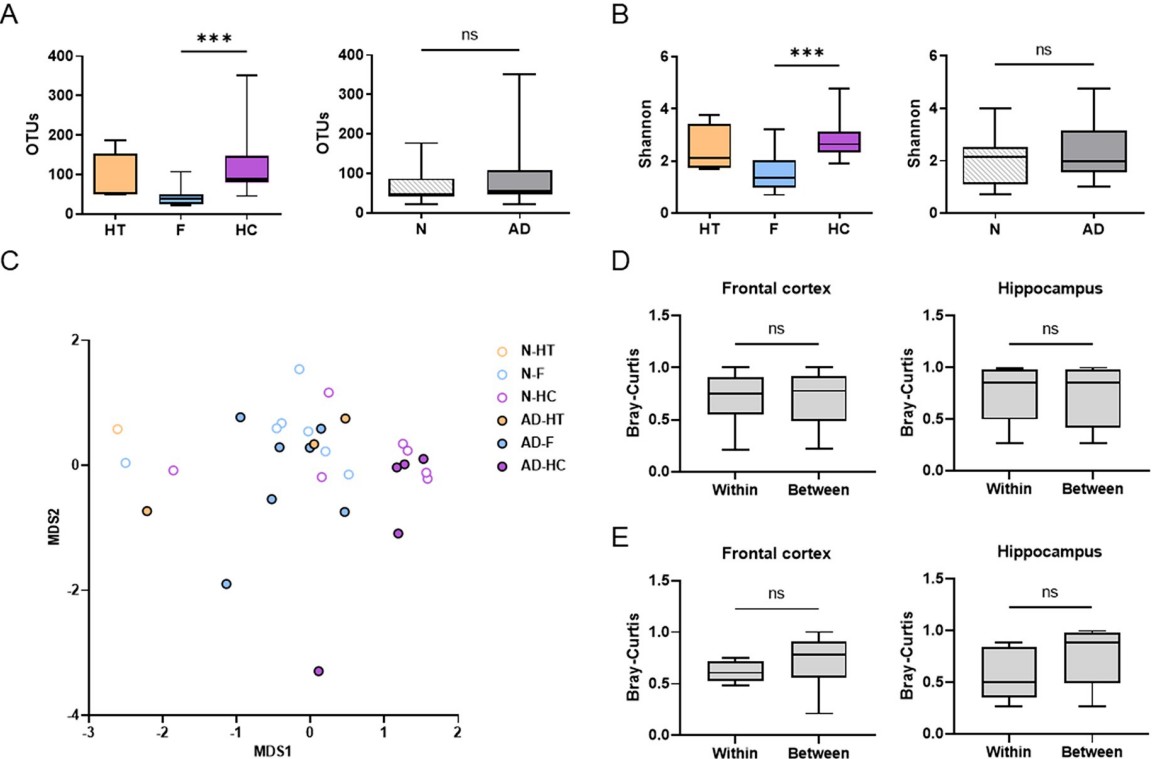

**Fig 2. Analysis of alpha- and beta-diversities in human brain tissues.** (A) OTUs of the HT, F, and HC, and those of N and AD are presented as box and whisker plots. (B) The Shannon indexes of the HT, F, and HC, and those of N and AD are presented as box and whisker plots (median: HT = 2.12, F = 1.34, HC = 2.64, N = 2.14, and AD = 2.03). (C) The multidimensional scaling (MDS) plot has been visualized using the Bray–Curtis and the significance of dissimilarities was examined by the permutational multivariate analysis of variance (HT vs F vs HC: $R^2$ = 0.19, $P$ = 0.001, N vs AD: $R^2$ = 0.03, $P$ = 0.647). (D) The Bray–Curtis distances of the N–N and AD–AD samples (Within), and those of N–AD samples (Between) were evaluated. (E) The Bray–Curtis distances of the (1)–(2) samples from the same donor (Within) and those of the samples from different donors (Between) were evaluated. The significance among the HT, F, and HC and between N and AD was examined by the Kruskal–Wallis test and Mann–Whitney $U$ test, respectively.

groups, regardless of whether the hypothalamus samples from AD patients were grouped as N or AD (Fig 2A, 2B and S1C, S1D Fig).

To determine the overall dissimilarities of each sample based on the brain region and disease, we used MDS plots based on the Bray–Curtis distance and the PERMANOVA (Fig 2C). The hippocampus, frontal cortex, and hypothalamus samples formed separate clusters ($R^2$ = 0.19, $P$ = 0.001), whereas there were no significant dissimilarities between the N and AD groups ($R^2$ = 0.03, $P$ = 0.647, when the AD–HT samples were grouped as AD; $R^2$ = 0.03, $P$ = 0.446, when the AD hypothalamus samples were grouped as N). We further examined the Bray–Curtis dissimilarity between the N and AD groups, specifically in the frontal cortex or hippocampus. The Bray–Curtis distances of the N–N and AD–AD samples (Within) were not different from those of the N–AD samples (Between) (Fig 2D).

To investigate the dissimilarities between two adjacent tissue blocks from the same donor, we examined the Bray–Curtis distances of the (1)–(2) samples from the same donor (Within) and those of the samples from different donors (Between). Although the differences were not significant, the Within distances tended to be smaller than the Between distances in both the frontal cortex and hippocampus, suggesting that the communities of adjacent blocks from the same donor shared more similarities (Fig 2E).

Collectively, bacteriotas in brain tissues were differentiated by the brain region rather than by AD.

## Taxonomic compositions of brain bacteriota

We conducted further investigations into the bacterial composition and relative abundance at the phylum, genus, and species levels. A total of 23 phyla were identified across all samples. The four most abundant phyla were Proteobacteria, Firmicutes, Actinobacteria, and Bacteroidetes, followed by Acidobacteria, Chloroflexi, Verrucomicrobia, and others. Proteobacteria was predominant, accounting for over 50% of the hypothalamus, frontal cortex, and hippocampus samples. While there were no significant differences in the phylum composition among the different brain regions, Proteobacteria showed a trend of being more abundant in the frontal cortex and hippocampus compared to the hypothalamus (statistically insignificant). Similarly, no significant difference was observed between the N and AD groups at the phylum level (Fig 3A and S1E Fig).

At the genus level, the top 20 genera accounted for 40% to 96% of the total bacteriota, and the top 50 genera accounted for 83% to 100% of the total bacteriota, with *Bradyrhizobium*, a member of Proteobacteria, being the most abundant in many samples (Fig 3B and S2 Table). Two genera presented significant differences in relative abundance across the different brain areas: KE159538_g (an uncharacterized genus belonging to Lachnospiraceae) was enriched in the hippocampus, while *Proteus*, which included only *Proteus mirabilis* species, was enriched in the hypothalamus (Fig 3C). Additionally, *Cloacibacterium*, which included only *Cloacibacterium normanense* species, was significantly enriched in the AD group ($P = 0.008$, when the AD hypothalamus samples were grouped as AD [Fig 3D]; $P = 0.001$, when the AD hypothalamus samples were grouped as N [S1F Fig]). *Stenotrophomonas* was significantly enriched in the AD group only when the AD hypothalamus samples were grouped as N ($P = 0.002$, S1F Fig). Among the species belong to *Stenotrophomonas*, *Stenotrophomonas maltophilia* group remained significantly enriched in the AD group when the AD hypothalamus samples were grouped as N ($P = 0.001$).

Interestingly, *Cloacibacterium normanense* was detected in the frontal cortex of all four AD patients and the hippocampus of two out of three AD patients, but not in any samples from control individuals or the hypothalamus of AD patients (Fig 3E). Although we expected to find numerous periodontal pathogens in the AD brain tissues, only *P. gingivalis* was detected in both N and AD groups without a statistical difference ($P = 0.888$; Fig 3F).

## Source compositions of brain bacteriota

To determine the origin of these bacteria, we investigated their source composition. We classified 710 species into 10 categories: oral, oral/gut, gut, skin/gut, skin, human others, environment, food, others, and unknown. The oral, gut, and skin groups consisted of bacteria resident in the oral cavity, gut, and skin, respectively. The human others category comprised bacteria isolated from clinical samples, blood, sputum, urogenital tracts, etc., whereas the environment group consisted of bacteria from soil, water, and air pollution. Bacteria from animals and shower hose were grouped as others (S2 Table).

The bacterial source composition in each brain tissue sample is presented in Fig 4A. Bacteria in the brain tissues mainly originated from human microbiome, with gut bacteria being dominant. The relative abundance of gut bacteria in the frontal cortex was significantly higher than that in the hippocampus. In contrast, human microbiome bacteria decreased in the hippocampus, and bacteria from environmental sources were significantly higher than in the frontal cortex (Fig 4B).

When the sources of bacteria detected in N and AD were compared, human others and others were significantly abundant in the AD group when the AD hypothalamus samples were grouped as AD (Fig 4C). The bacteria uniquely detected in the AD brain tissues and

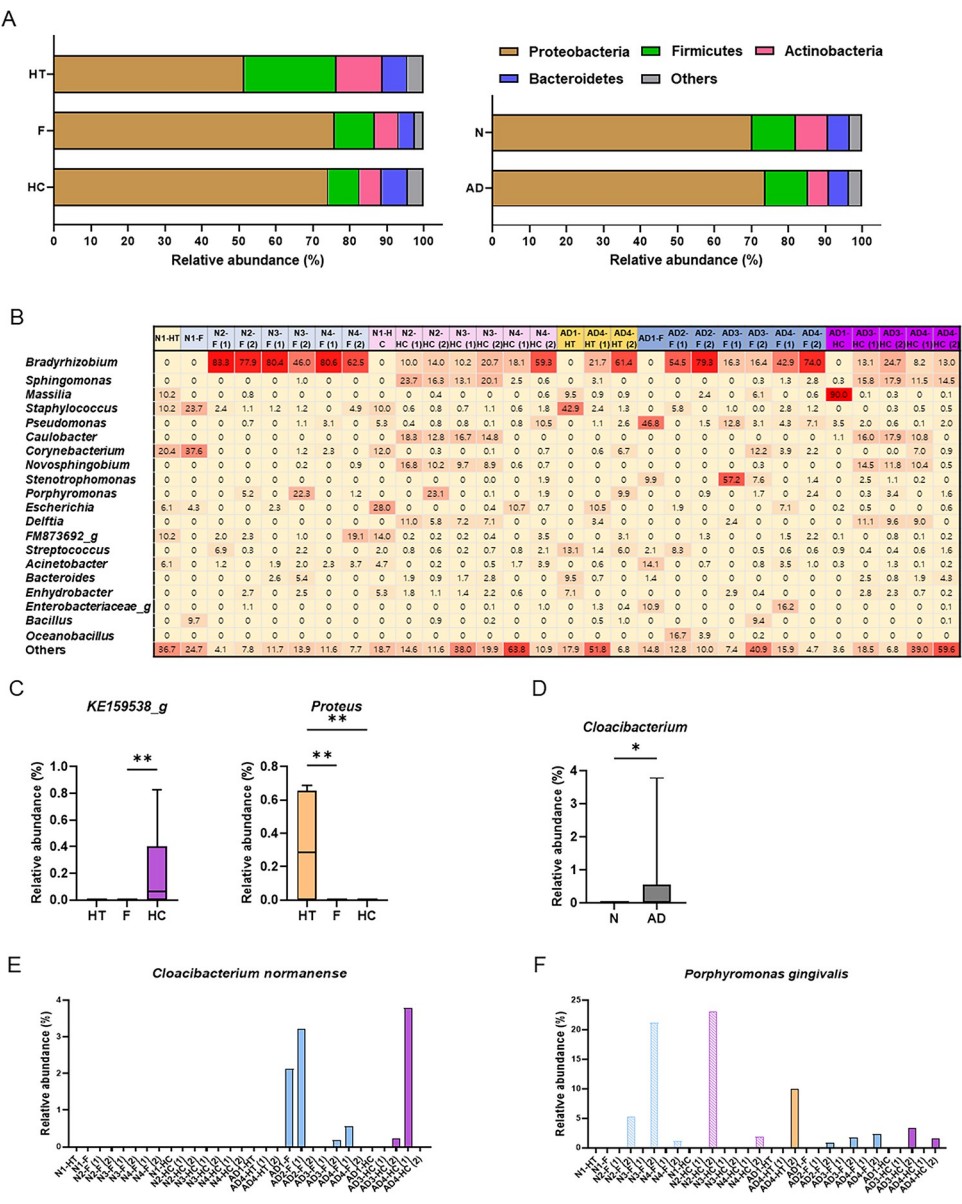

**Fig 3. Bacteria composition analysis in human brain tissues at the phylum, genus, and species levels.** (A) Compositions of phyla from brain samples are presented. (B) Heatmap of top 20 genera. The numbers indicate relative abundance. (C) Relative abundance of the genus across distinct brain areas was examined and the significantly enriched genera, KE159538_g and *Proteus*, are presented. (D) Relative abundance of the genus between N and AD was examined and the significantly enriched in AD, *Cloacibacterium*, is presented. (E, F) Rela G tive abundance of the species was analyzed between N and AD. The relative abundance of (E) *Cloacibacterium normanense* and (F) *Porphyromonas gingivalis* across the samples are presented. The significance among the HT, F, and HC and between N and AD was examined by the Kruskal–Wallis test and Mann–Whitney *U* test, respectively.

categorized as human pathogens included *Anaerococcus tetradius*, *Pseudomonas stutzeri*, *Massilia consociate*, *Massilia timonae*, *Nocardia nova*, and *Helicobacter pullorum*. Importantly, bacteria originated from human others or others were not detected in negative controls, limiting the possibility that these differences may be attributed to contamination (Fig 4A).

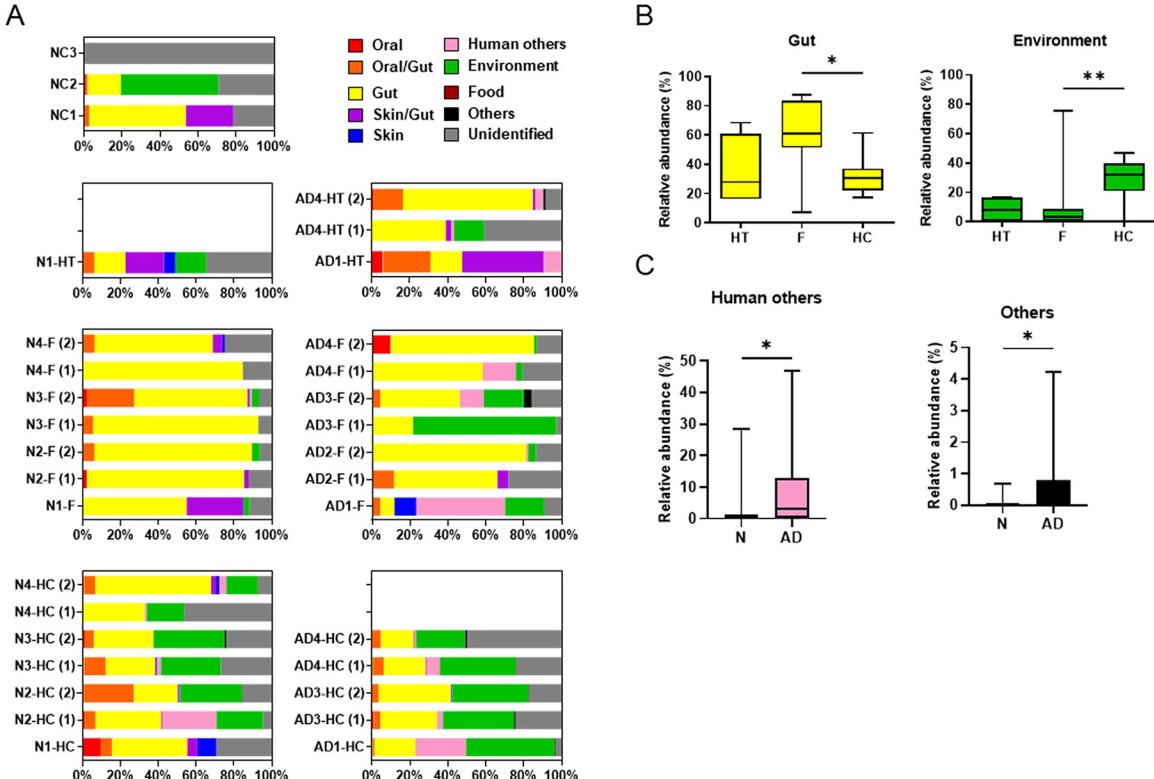

**Fig 4. Bacterial source composition in human brain tissues.** (A) Bacterial source compositions of all species in the samples are presented. (B) Relative abundance of significantly enriched bacterial sources in the distinct tissues and (C) that of significantly enriched bacterial sources in AD are presented. The significance among the HT, F, and HC and between N and AD was examined by Kruskal–Wallis test and Mann–Whitney *U* test, respectively.

## Discussion

To investigate potential associations between specific bacteria and AD pathology, we analyzed 30 postmortem brain tissue samples from four controls and four AD patients, including both AD-associated neurodegenerative regions (frontal cortex and hippocampus) and the hypothalamus as a non-AD-associated region. Our findings reveal no significant differentiation in the low-biomass brain bacteriota attributed to the presence of AD pathology, except for the distinctive detection of *C. normanense* and the notable enrichment of infection-associated bacteria.

In low-biomass microbiome studies, the elimination of off-target amplicons and potential contaminants is crucial. A recent investigation involving human brain tissues (olfactory bulb and pre-frontal cortex) from healthy subjects and patients with Parkinson's disease highlighted off-target amplification as a major confounding factor in studies characterized by low bacterial but high host DNA content [21]. In that study, mapping against human reference genome using minimap2 removed 34.2% OTUs as off-target products, but seemingly brain-enriched microbes turned out to be human DNA sequences in further analysis, leading to a conclusion that evidence for the presence of a brain bacteriota is absent. In contrast, in the current study, an average of 96.6% of total reads were removed as off-target amplicons by utilizing the nhmmer program. This rigorous filtering ensured free of off-target issues, and most reads annotated as known species presented > 99% sequence homology. We treated reads with low abundance (≤2) or those consistently distributed across the three NC and 30 brain tissue

samples as potential contaminants. Although some studies eliminate all taxa detected in NC samples, this approach may lead to false-negatives. For example, *Acidovorax*, *Acinetobacter*, *Corynebacterium*, *Deinococcus*, and *Streptococcus* are reported as contaminant genera detected in the sequencing of negative blank controls but also identified in the blood of healthy humans after thorough filtering for contaminants [22, 23].

The first characteristic of brain bacteriota observed in the current study was an extremely low biomass. All studies, including the current one, amplified more than 30 cycles to sequence 16S rRNA gene and used a second PCR consisting of a total of 69–80 cycles to detect specific bacterial genes within human brain tissues [15, 17–20]. This indicates the inherent challenges in amplifying bacterial DNA from such tissues. In our previous study, intratissue bacteriota in human gingival tissues with periodontitis were examined using the same sequencing and bio-informatic platforms, in which more than 17,000 valid reads per sample were obtained without a second PCR [24]. In this study, however, less than 1,000 valid reads were acquired in half of the brain tissue samples, despite the additional amplification with eight cycles.

Interestingly, the number of valid reads detected in the hippocampus was significantly higher than those in the hypothalamus and frontal cortex (Fig 1B). This might be attributed to the unique physiological characteristics of the hippocampus, that is, its heightened BBB perme-ability in comparison with other brain regions [25]. Although estimation of the true microbial biomass usually entails qPCR, the presence of more than 85% off-target amplicons in our study suggests that our qPCR results (Fig 1C) may not accurately reflect the genuine microbial biomass. A multitude of factors, including library quantitation for pooling, off-target amplifi-cation, and exogenous contamination, can affect the number of valid reads obtained from 16S rRNA sequencing. Notably, hippocampal samples exhibited higher total read numbers but lower proportions of off-target amplicons compared to the hypothalamus and frontal cortex (S2 Fig), which raises the possibility that the higher valid reads observed in the hippocampus might stem from experimental artifacts.

The second intriguing finding of our study was that the overall structure of the brain bacter-iota varied by brain regions rather than by the presence of AD (Fig 2C). However, this needs replication in future studies because the small number of specimens used in the current study limited the power to detect statistical significance. Although most analyzed parameters revealed no distinction in the brain bacteriota between the N and AD groups, *C. normanense* was uniquely detected in AD-associated neurodegenerative regions (Fig 3E). *C. normanense* is a Gram-negative, non-motile bacterium commonly found in human gut and wastewater [20, 26]. Its role in host interaction remains uncharacterized, making it a focal point for future research on its potential involvement in AD pathogenesis. *Cutibacterium* (formerly *Propioni-bacterium*) *acnes*, a species previously suggested as a potential contributor to neuroinflamma-tion [18], was identified in the initial dataset, reaching up to 5.5% (0 to 42 reads per sample), but was subsequently excluded as a contaminant in the filter 3. Emery et al. [18] interpreted the elevated bacterial reads (both total and *C. acnes*) in AD brains as indicative of increased bacterial levels. However, as mentioned earlier, read counts do not accurately represent the true microbial biomass. The predominance of *C. acnes* in AD brains was not replicated in a subsequent study by the same group [19], though differences in the brain regions studied (tem-poral cortex vs. hippocampus and locus coeruleus) could account for this inconsistency.

Intriguingly, when investigating the known sources of identified taxa, more than 50% of the bacteriota in 16 of the 30 brain samples belonged to the human microbiome members (Fig 4A). Nevertheless, Proteobacteria (i.e., *Bradyrhizobium*) overwhelmingly dominated the bacteriota in all brain samples (Fig 3A). This pattern, differing from the phylum composition of gut, oral cavity, or skin bacteriotas but resembling that of blood bacteriota [23, 27], hints at the possible entry of circulating bacteria or bacterial components containing DNA into brain

tissues. *Bradyrhizobium japonicum*, the sole species identified within the *Bradyrhizobium* genus, is known for its symbiotic relationship with legume roots but has also been identified in the human colon [28]. Significantly, its association with prostate cancer, lung cancer, and colorectal cancer has been documented [28–30].

Furthermore, the absence of differences in phylum composition between the N and AD groups (Fig 3A and S1E Fig) suggests that bacterial translocation into brain tissue might be linked to increased BBB permeability due to inflammation, rather than neuroinflammation itself. This notion is supported by the significant enrichment of infection-associated bacteria from human clinical samples (classified as human others) in the AD brains (Fig 4C). Notably, most of these bacteria have connections to infectious diseases such as sepsis, genital ulcer disease, peritonitis, brain abscess, and septic abortion [31–38]. In fact, any common systemic infections (sepsis, pneumonia, other lower respiratory tract infections, urinary tract infections, and skin and soft tissue infections) are known to elevate the risk of dementia diagnosis, as evidenced in a population-based cohort study [39]. Moreover, any systemic inflammation conditions, including surgery and aseptic insults, are linked to AD [19]. Circulating inflammatory cytokines in blood can cause breakdown in BBB, allowing for influx of inflammatory mediators and immune cells into the brain parenchyma that result in subsequent neuroinflammation [40]. The enrichment of infection-associated bacteria in AD brains could imply two scenarios: either these bacteria have a direct role in neuroinflammation compared to bacteria from the human microbiome or their presence might reflect BBB disruption during infection. When the AD hypothalamus samples were grouped as N, the infection-associated bacteria were not significantly enriched in AD (S1G Fig), thereby supporting the second scenario more than the first.

Contrary to our anticipation, we failed to find substantial evidence linking AD to periodontal pathogens. While *P. gingivalis* was detected in the brains of both N and AD groups (Fig 3E), no other periodontal pathogens were identified. Despite some studies reporting the preferential detection of *P. gingivalis* LPS or gingipains in AD brain tissues compared to controls [14, 15], the 16S rRNA gene was prevalently detected in both control and AD brain tissues in the same study [15]. Immunofluorescence findings by Pool et al. also indicated the absence of other periodontal pathogens, namely, *Treponema denticola* and *Tannerella forsythia*, in both control and AD brain tissues [14]. These coincide with the facts that *P. gingivalis*, but not either *T. denticola* or *T. forsythia*, is enriched within the gingival tissue of periodontitis patients and identified as blood bacteriota [23, 24]. Although the gingipain inhibitor COR388 successfully prevented *P. gingivalis*-induced neuroinflammation and neurotoxicity in a mouse model, its clinical trial failed to show statistically significant efficacy [41]. Therefore, the epidemiological association between periodontitis and AD might be mediated by increased inflammatory mediators in circulation rather than the direct involvement of periodontal pathogens in the brain.

Several potential pathways for bacterial translocation into the brain have been proposed, including bacterial breach through the transcytosis of endothelial cells in the BBB or transportation across the BBB within infected immune cells [42–46]. However, whether live bacteria or bacterial components are present in the brain remains uncertain. Previous research demonstrated the transient translocation of bacterial extracellular vesicles (EVs) into the brain within an hour after intraperitoneal injection in mice [47]. To explore this possibility, we quantified the DNA content of *P. gingivalis* EVs using 16S rRNA gene qPCR. Remarkably, 1 μg of *P. gingivalis* EVs, secreted from approximately $1.5 \times 10^{11}$ cells, contained only 64 DNA copies. If the bacterial DNA detected in brain tissues originates from EV particles rather than from whole cells, the observed low biomass in our study might correspond to a much higher presence of bacterial components [48, 49]. This underlines the need for further exploration into the role of EVs in bacterial translocation to the brain.

In conclusion, the low-biomass brain bacteriota exhibited only subtle changes due to AD pathology, raising new questions on the role of bacterial infection in neuroinflammation.

## Supporting information

**S1 Fig. Microbiome analysis in normal and AD patients when the hypothalamus samples from AD patients were grouped as N.** (A) Valid reads, (B) copy number, (C) OTUs, and (D) the Shannon index of N and AD are presented as box and whisker plots. (E) Compositions of phyla from brain samples are presented. (F) Relative abundance of the significantly enriched genus in N or AD is presented. (G) Relative abundance of the significantly enriched bacterial sources in N or AD is presented. The significance between N and AD was examined by the Mann–Whitney U test.
(TIF)

**S2 Fig. Potential contaminants ratio from total reads in human brain tissues.** Total reads, non-specific amplicon/total reads, and contaminants/total reads in different brain areas are depicted. The significance among the HT, F, and HC was examined by the Kruskal–Wallis test.
(TIF)

**S1 Table. Relative abundance of Top 50 genera in the brain areas of each subject type.**
(DOCX)

**S2 Table. Classification of species sources.**
(DOCX)

## Acknowledgments

The brain research resource (human brain tissue) was provided by Korea Brain Bank Network [Asan Medical Center Brain Bank (Eun-Jae Lee, Soo Jeong Nam), Kangwon National University Hospital Brain Bank (Seongheon Kim, Yeshin Kim), Severance Brain Bank (Sohn Young Ho, Oh Ji Woong)] operated by the Korea Brain Research Institute and Ministry of Science and ICT.

## Author Contributions

**Conceptualization:** Se-Young Choi, Youngnim Choi.

**Data curation:** Yeon Kyeong Ko.

**Formal analysis:** Yeon Kyeong Ko, Eunbi Kim.

**Funding acquisition:** Youngnim Choi.

**Investigation:** Yeon Kyeong Ko.

**Methodology:** Yeon Kyeong Ko, Eunbi Kim.

**Project administration:** Youngnim Choi.

**Resources:** Eun-Jae Lee, Soo Jeong Nam, Yeshin Kim, Seongheon Kim, Hyun Young Kim.

**Supervision:** Youngnim Choi.

**Visualization:** Yeon Kyeong Ko, Eunbi Kim.

**Writing – original draft:** Yeon Kyeong Ko.

**Writing – review & editing:** Eunbi Kim, Eun-Jae Lee, Soo Jeong Nam, Yeshin Kim, Seongheon Kim, Se-Young Choi, Hyun Young Kim, Youngnim Choi.

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
