## [Decision Letter · Decision Letter 0]

20 Nov 2023

PONE-D-23-27926Enrichment of infection-associated bacteria in the low biomass brain bacteriota of Alzheimer’s disease patients

PLOS ONE

Dear Dr. Choi,

Thank you for submitting your manuscript to PLOS ONE. After careful consideration by 2 Reviewers and an Academic Editor, all of the critiques of both Reviewers must be addressed in detail in a revision to determine publication status. If you are prepared to undertake the work required, I would be pleased to reconsider my decision, but revision of the original submission without directly addressing the critiques of the Reviewers does not guarantee acceptance for publication in PLOS ONE. If the authors do not feel that the queries can be addressed, please consider submitting to another publication medium. A revised submission will be sent out for re-review. The authors are urged to have the manuscript given a hard copyedit for syntax and grammar.

We look forward to receiving your revised manuscript.

Kind regards,

Stephen D. Ginsberg, Ph.D.

Section Editor

PLOS ONE

Journal Requirements:

"This study was supported by the National Research Foundation of Korea (Daejun, Korea) through the grants 2018R1A5A2024418 and 2020R1A2C2007038 awarded to Youngnim Choi. The funders had no role in study design, data collection and analysis, decision to publish, or

preparation of the manuscript."

"This study was supported by the National Research Foundation of Korea (Daejun, Korea) through the grants 2018R1A5A2024418 and 2020R1A2C2007038 awarded to Youngnim Choi. The funders had no role in study design, data collection and analysis, decision to publish, or

preparation of the manuscript."

Reviewers' comments:

**Comments to the Author**

1. Is the manuscript technically sound, and do the data support the conclusions?

Reviewer #1: Yes

Reviewer #2: Partly

2. Has the statistical analysis been performed appropriately and rigorously? 

Reviewer #1: Yes

Reviewer #2: Yes

3. Have the authors made all data underlying the findings in their manuscript fully available?

Reviewer #1: Yes

Reviewer #2: No

4. Is the manuscript presented in an intelligible fashion and written in standard English?

Reviewer #1: Yes

Reviewer #2: Yes

5. Review Comments to the Author

Reviewer #1: The authors conducted 16S rRNA gene sequencing of 30 postmortem brain tissues from four individuals with normal and four AD patients. As the authors acknowledged, no specific bacterial species associated with AD pathology were identified that the authors had expected at the beginning of the study. However, the study is still significant on the point that some of the characteristics of the bacteria existing in AD and healthy brains were revealed. The quality of the paper would be improved if the following points were additionally mentioned.

1. Please comment as a limitation in the Discussion that the power to detect statistical significance is insufficient due to the small number of specimens.

2. Note that the fact that no significant difference does not suggest that there was no difference. For example, in the second paragraph of Overall structures of brain bacteria, “suggesting no overall differences in the bacterial communities of either frontal cortex or hippocampus between the N and AD groups”. This description may be misleading to readers and should be removed.

3. Regarding classification 710 bacterial species into 10 categories in the Fig 4, I am not familiar with such classification. Are these classifications generally accepted? If so, please cite previous literatures related this classification. On the other hand, if the classification is made by the authors themselves, please submit the classification list as a Supplementary Table for future researchers to refer to.

Reviewer #2: General Comment. This is an interesting and well written paper about possible differences between the microbiota of Alzheimer’s and normal post-mortem brains. The authors report not finding any clear differences between these types of subjects in the areas of the brain sampled. They also report that their findings differ from what other researchers have found. My comments and questions are primarily focused on whether the author's methodology could account for the different findings and suggesting additional data details that enable a deeper understanding of the results. Specific questions and comments follow.

Contamination

The authors suggest that contamination of other researcher’s results could be an issue and specifically state that others have not sequenced any blank controls. One of these researchers in reference 18 specifically addresses the issue of contamination which we quote below, indicating that even if there were blank controls that showed contamination that they would not have been able to explain the large observed differences between AD and normal subjects, specifically Cutibacterium acnes.

“However, the consistently high levels seen here in AD samples compared to normal brains and the apparent minimal contribution of post mortem interval along with the lack of significant contact with skin makes contamination an unlikely explanation for the P. acnes content of these data.”

1) Could the authors explain why they think other researchers lack of blank controls seriously affected their results in comparison to the authors' new results.

Bacteria Identification.

1) In reference 18, Emery et al. identify C. acnes (formerly P. acnes) as a bacterium whose abundance is very different between AD and N subjects. Could the authors explain why they do not report observing C. acnes?

2) Were there differences in the author’s sequencing protocol that could account for this? Could Cutibacterium (Propionibacterium) have been misidentified as something else? Here is a reference that addresses this somewhat.

Jacquelyn S. Meisel, Geoffrey D. Hannigan, Amanda S. Tyldsley, Adam J. SanMiguel, Brendan P. Hodkinson, Qi Zheng, Elizabeth A. Grice, Skin Microbiome Surveys Are Strongly Influenced by Experimental Design, Journal of Investigative Dermatology, Volume 136, Issue 5, 2016, Pages 947-956, ISSN 0022-202X,

https://doi.org/10.1016/j.jid.2016.01.016.,

link to useful supplemental table from this reference: https://ars.els-cdn.com/content/image/1-s2.0-S0022202X16003699-mmc1.pdf

I suggest that the authors discuss the above issues in a paragraph.

3) I understand that community statistical tests did not detect any significant differences between the various groupings. On the other hand, I did not see any lists of the main bacteria (genus level abundances) detected (except for a few examples) that would enable the reader to have a sense what was underlying these statistical computations. I suggest that the major bacteria detected be tabulated for the subjects/samples.

Bacterial Sources.

1) It would be useful to see tables of major bacteria (genus level abundances) that compare the brain areas of each subject type and the source locations for comparison purposes.

6. PLOS authors have the option to publish the peer review history of their article (what does this mean?). If published, this will include your full peer review and any attached files.

**Do you want your identity to be public for this peer review?** For information about this choice, including consent withdrawal, please see our Privacy Policy.

Reviewer #1: No

Reviewer #2: **Yes: **Jeffrey R. Lapides

---

## [Author Response · Author response to Decision Letter 0]

29 Nov 2023

We appreciate the reviewers for their time and constructive comments on our manuscript.

Reviewer #1: The authors conducted 16S rRNA gene sequencing of 30 postmortem brain tissues from four individuals with normal and four AD patients. As the authors acknowledged, no specific bacterial species associated with AD pathology were identified that the authors had expected at the beginning of the study. However, the study is still significant on the point that some of the characteristics of the bacteria existing in AD and healthy brains were revealed. The quality of the paper would be improved if the following points were additionally mentioned.

1. Please comment as a limitation in the Discussion that the power to detect statistical significance is insufficient due to.

→ It has been added in the Discussion section as following (lines 372-374): However, this needs replication in future studies because the small number of specimens used in the current study limited the power to detect statistical significance.

2. Note that the fact that no significant difference does not suggest that there was no difference. For example, in the second paragraph of Overall structures of brain bacteria, “suggesting no overall differences in the bacterial communities of either frontal cortex or hippocampus between the N and AD groups”. This description may be misleading to readers and should be removed.

→ Removed.

3. Regarding classification 710 bacterial species into 10 categories in the Fig 4, I am not familiar with such classification. Are these classifications generally accepted? If so, please cite previous literatures related this classification. On the other hand, if the classification is made by the authors themselves, please submit the classification list as a Supplementary Table for future researchers to refer to.

→ The list has been provided as Supplementary Table 2. In its process, we discovered that the most abundant species Bradyrhizobium japonicum has been misclassified as oral/gut from gut, resulting in changes in Figs 4A and B.

Reviewer #2: General Comment. This is an interesting and well written paper about possible differences between the microbiota of Alzheimer’s and normal post-mortem brains. The authors report not finding any clear differences between these types of subjects in the areas of the brain sampled. They also report that their findings differ from what other researchers have found. My comments and questions are primarily focused on whether the author's methodology could account for the different findings and suggesting additional data details that enable a deeper understanding of the results. Specific questions and comments follow.

Contamination

The authors suggest that contamination of other researcher’s results could be an issue and specifically state that others have not sequenced any blank controls. One of these researchers in reference 18 specifically addresses the issue of contamination which we quote below, indicating that even if there were blank controls that showed contamination that they would not have been able to explain the large observed differences between AD and normal subjects, specifically Cutibacterium acnes.

“However, the consistently high levels seen here in AD samples compared to normal brains and the apparent minimal contribution of post mortem interval along with the lack of significant contact with skin makes contamination an unlikely explanation for the P. acnes content of these data.”

1) Could the authors explain why they think other researchers lack of blank controls seriously affected their results in comparison to the authors' new results.

→ C. acnes was identified in the initial data set of our study but filtered out in the exact binomial test in comparison with the three negative controls. Furthermore, the author of ref 18 reported increased levels of C. acnes based on the elevated read counts, which do not accurately represent the true microbial biomass. We discussed this in lines 379-388 of Discussion section as following: Cutibacterium (formerly Propionibacterium) acnes, a species previously suggested as a potential contributor to neuroinflammation (18), was identified in the initial dataset, reaching up to 5.5% (0 to 42 reads per sample), but was subsequently excluded as a contaminant in the filter 3. Emery et al. (18) interpreted the elevated bacterial reads (both total and C. acnes) in AD brains as indicative of increased bacterial levels. However, as mentioned earlier, read counts do not accurately represent the true microbial biomass. The predominance of C. acnes in AD brains was not replicated in a subsequent study by the same group (19), though differences in the brain regions studied (temporal cortex vs. hippocampus and locus coeruleus) could account for this inconsistency.

Bacteria Identification.

1) In reference 18, Emery et al. identify C. acnes (formerly P. acnes) as a bacterium whose abundance is very different between AD and N subjects. Could the authors explain why they do not report observing C. acnes?

2) Were there differences in the author’s sequencing protocol that could account for this? Could Cutibacterium (Propionibacterium) have been misidentified as something else? Here is a reference that addresses this somewhat.

Jacquelyn S. Meisel, Geoffrey D. Hannigan, Amanda S. Tyldsley, Adam J. SanMiguel, Brendan P. Hodkinson, Qi Zheng, Elizabeth A. Grice, Skin Microbiome Surveys Are Strongly Influenced by Experimental Design, Journal of Investigative Dermatology, Volume 136, Issue 5, 2016, Pages 947-956, ISSN 0022-202X,

https://doi.org/10.1016/j.jid.2016.01.016.,

link to useful supplemental table from this reference: https://ars.els-cdn.com/content/image/1-s2.0-S0022202X16003699-mmc1.pdf

I suggest that the authors discuss the above issues in a paragraph.

→ Thank you for providing a useful reference. According to the reference paper, Propionibacterium is underrepresented in V4 datasets compared to V1-V3 datasets. Because Emory et al. (ref 18) used primers targeting V3, and we used primers targeting V3-V4, the discrepancy is not likely due to a difference in the sequencing protocol. As described above, we discussed i) contamination issue, ii) read counts as insufficient indicator of biomass, and iii) variation in brain regions studied.

3) I understand that community statistical tests did not detect any significant differences between the various groupings. On the other hand, I did not see any lists of the main bacteria (genus level abundances) detected (except for a few examples) that would enable the reader to have a sense what was underlying these statistical computations. I suggest that the major bacteria detected be tabulated for the subjects/samples.

→ We included a heatmap for top 20 genera in each sample (Fig 3B) and the mean relative abundance of for top 50 genera in the brain areas of each subject type (Supplementary Table 1). The new result was described in the Result section (lines 275-277) and the most abundant genus Bradyrhizobium was discussed in the Discussion section (lines 395-398) as following:

[Result] At the genus level, the top 20 genera accounted for 40% to 96% of the total bacteriota, and the top 50 genera accounted for 83% to 100% of the total bacteriota, with Bradyrhizobium, a member of Proteobacteria, being the most abundant in many samples (Fig 3B and Table S2).

[Discussion] Bradyrhizobium japonicum, the sole species identified within the Bradyrhizobium genus, is known for its symbiotic relationship with legume roots but has also been identified in the human colon (28). Significantly, its association with prostate cancer, lung cancer, and colorectal cancer has been documented (28-30).

Bacterial Sources.

1) It would be useful to see tables of major bacteria (genus level abundances) that compare the brain areas of each subject type and the source locations for comparison purposes.

→ In Supplementary Table 2, the species classified into each bacterial source is listed.

---

## [Decision Letter · Decision Letter 1]

10 Dec 2023

Enrichment of infection-associated bacteria in the low biomass brain bacteriota of Alzheimer’s disease patients

PONE-D-23-27926R1

Dear Dr. Choi,

We’re pleased to inform you that your manuscript has been judged scientifically suitable for publication and will be formally accepted for publication once it meets all outstanding technical requirements.

Kind regards,

Stephen D. Ginsberg, Ph.D.

Section Editor

PLOS ONE

**Comments to the Author**

1. If the authors have adequately addressed your comments raised in a previous round of review and you feel that this manuscript is now acceptable for publication, you may indicate that here to bypass the “Comments to the Author” section, enter your conflict of interest statement in the “Confidential to Editor” section, and submit your "Accept" recommendation.

Reviewer #1: All comments have been addressed

Reviewer #2: All comments have been addressed

2. Is the manuscript technically sound, and do the data support the conclusions?

Reviewer #1: Yes

Reviewer #2: Yes

3. Has the statistical analysis been performed appropriately and rigorously? 

Reviewer #1: Yes

Reviewer #2: Yes

4. Have the authors made all data underlying the findings in their manuscript fully available?

Reviewer #1: Yes

Reviewer #2: Yes

5. Is the manuscript presented in an intelligible fashion and written in standard English?

Reviewer #1: Yes

Reviewer #2: Yes

6. Review Comments to the Author

Reviewer #1: I am appreciate for your revising the manuscript.

All comments have been addressed. I could not find any other problem.

Reviewer #2: (No Response)

7. PLOS authors have the option to publish the peer review history of their article (what does this mean?). If published, this will include your full peer review and any attached files.

Reviewer #1: No

Reviewer #2: No

---

## [Editor Report · Acceptance letter]

27 Dec 2023

PONE-D-23-27926R1 

PLOS ONE

Dear Dr. Choi, 

I'm pleased to inform you that your manuscript has been deemed suitable for publication in PLOS ONE. Congratulations! Your manuscript is now being handed over to our production team.

Kind regards, 

on behalf of

Dr. Stephen D. Ginsberg 

Section Editor

PLOS ONE